# A heterogeneously integrated lithium niobate-on-silicon nitride photonic platform

Mikhail Churaev [1,2,5], Rui Ning Wang [1,2,5], Annina Riedhauser[3,5], Viacheslav Snigirev[1,2], Terence Blésin [1,2], Charles Möhl [3], Miles H. Anderson[1,2], Anat Siddharth [1,2], Youri Popoff[3,4], Ute Drechsler[3], Daniele Caimi[3], Simon Hönl [3], Johann Riemensberger [1,2], Junqiu Liu[1,2], Paul Seidler [3] ✉ & Tobias J. Kippenberg [1,2] ✉

The availability of thin-film lithium niobate on insulator (LNOI) and advances in processing have led to the emergence of fully integrated LiNbO$_3$ electro-optic devices. Yet to date, LiNbO$_3$ photonic integrated circuits have mostly been fabricated using non-standard etching techniques and partially etched waveguides, that lack the reproducibility achieved in silicon photonics. Widespread application of thin-film LiNbO$_3$ requires a reliable solution with precise lithographic control. Here we demonstrate a heterogeneously integrated LiNbO$_3$ photonic platform employing wafer-scale bonding of thin-film LiNbO$_3$ to silicon nitride (Si$_3$N$_4$) photonic integrated circuits. The platform maintains the low propagation loss (<0.1 dB/cm) and efficient fiber-to-chip coupling (<2.5 dB per facet) of the Si$_3$N$_4$ waveguides and provides a link between passive Si$_3$N$_4$ circuits and electro-optic components with adiabatic mode converters experiencing insertion losses below 0.1 dB. Using this approach we demonstrate several key applications, thus providing a scalable, foundry-ready solution to complex LiNbO$_3$ integrated photonic circuits.

Modern society has a constantly increasing demand for optical communications bandwidth, with aggregate data rates doubling every 18 months[1], and optical modulators play a crucial role in this context, providing the means to transfer electronic signals to optical carriers. With the rise of commercial integrated photonics, a wide variety of modulation platforms have been demonstrated that are compatible with wafer-scale manufacturing, among which silicon and indium phosphide are the most prominent[2–5]. Among all the materials used, lithium niobate (LiNbO$_3$) remains unique in term of its excellent physical properties and commercial availability[6,7]. Advances in wafer-scale transfer of LiNbO$_3$ thin-films via the SmartCut™ technique, combined with improvements in its etching, have enabled low-loss integrated photonic circuits based on thin film lithium niobate[8–11]. This has led to several key demonstrations, including ultra-high-Q optical microresonators[9], efficient electro-optic frequency comb generation[12], frequency converters[13] and non-reciprocal devices[14,15]. In addition, using LiNbO$_3$integrated photonic circuits, electro-optic modulation both at CMOS voltage levels and at high speed (beyond 100 GHz) has been achieved[16,17], offering routes toward compact integrated LiNbO$_3$ modulators compatible for applications ranging from classical communication for 5G cellular networks and data-center interconnects to quantum interfaces for microwave-optical conversion[18–20], and topological photonics employing synthetic dimensions[21,22]. Besides the electro-optic applications, integrated LiNbO$_3$ photonic integrated circuits (PICs) are also of high interest for nonlinear photonics, for example, for efficient second-harmonic generation, optical squeezing, parametric amplification, and Kerr comb generation[23–26].

[1]Institute of Physics, Swiss Federal Institute of Technology Lausanne (EPFL), CH-1015 Lausanne, Switzerland. [2]Center for Quantum Science and Engineering, EPFL, Lausanne, Switzerland. [3]IBM Research - Europe, Zurich, Säumerstrasse 4, CH-8803 Rüschlikon, Switzerland. [4]Integrated Systems Laboratory, Swiss Federal Institute of Technology Zurich (ETH Zürich), CH-8092 Zürich, Switzerland. [5]These authors contributed equally: Mikhail Churaev, Rui Ning Wang, Annina Riedhauser. ✉e-mail: pfs@zurich.ibm.com; tobias.kippenberg@epfl.ch

Despite numereous advances to date, widespread adoption of LiNbO$_3$ integrated photonics is still impeded by several key issues. First, current LNOI-based devices are fabricated using specific non-conventional (i.e. non-Si-based-foundry) ion-beam etching (IBE) to achieve smooth waveguide surfaces. Insufficient etch-mask selectivity leads to the formation of shallow ridge waveguides that require more challenging process control to achieve the desired geometries, compared to the standard strip waveguides that are used in commercial Si or Si$_3$N$_4$ PIC foundries. This complicates the establishment of a reliable process design kit (PDK) for integrated LiNbO$_3$ platforms. Second, edge coupling between fibers and chips is challenging, as the ridge waveguide structures demonstrated so far show significant coupling loss, 5 to 10 dB per facet[13], unless more complicated double-etching techniques are used[27,28]. Third, while record resonance quality factors (Q ≈ 10$^7$, linear loss of 2.7 dB/m) have been reported in LiNbO$_3$ microresonators[9], this has only been demonstrated for selected optical resonances and has not been achieved broadly in other recently reported works, where losses are typically one order of magnitude higher (20–30 dB/m, see Supplementary Table 1 for comparison). For future applications, uniformly low loss across a wafer using precise and mature lithographic processes, along with efficient coupling, are necessary to develop a foundry-level technology that includes PDKs with, e.g., splitters, arrayed-waveguide gratings, optical filters or beamforming networks.

As an alternative to conventional bulk LiNbO$_3$ and ridge-waveguide-based photonic devices, hybrid platforms combining thin-film LiNbO$_3$ with waveguides made of Si, Si$_3$N$_4$, or Ta$_2$O$_5$ have been recently developed[29–31] (see Fig. 1a, b). With proper geometry optimization, the heterogeneously integrated LNOI devices can reach electro-optic performance comparable to that of the all-LNOI platforms[32] ($V_\pi L$ = 2.3 V·cm for push-pull Mach-Zehnder modulators, which corresponds to 4.6 V·cm phase modulation). Heterogeneous integration using the organic adhesive benzocyclobutene (BCB) to bond LNOI to silicon and direct bonding of chiplets to silicon and silicon nitride PICs has been demonstrated, leading to modulators operating at CMOS voltages[30,33]. However, wafer-level bonding has to date never been shown. Even though hybrid Si$_3$N$_4$-LiNbO$_3$ waveguides were reported[34,35], the previous works employed only short (mm scale) chiplets with strong transition losses to the chiplets, compounding precise determination of the losses. All prior approaches were aimed at specific device applications and could not reveal numerous benefits of heterogeneous integration of LiNbO$_3$ with a well-developed photonic platform.

In this work, we demonstrate a high-yield, low-loss, integrated LiNbO$_3$-Si$_3$N$_4$ photonic platform that solves multiple issues of LNOI integrated photonics. This is achieved by wafer-scale heterogeneous integration[34,36] (i.e. direct wafer bonding[37]) of an LNOI wafer onto a patterned and planarized ultra-low-loss Si$_3$N$_4$ substrate as depicted in Fig. 1c. Compared to the die-level integration, the wafer-scale bonding has several distinct advantages, chief among them an increased wafer throughput that is advantageous in high volume applications[38]. In

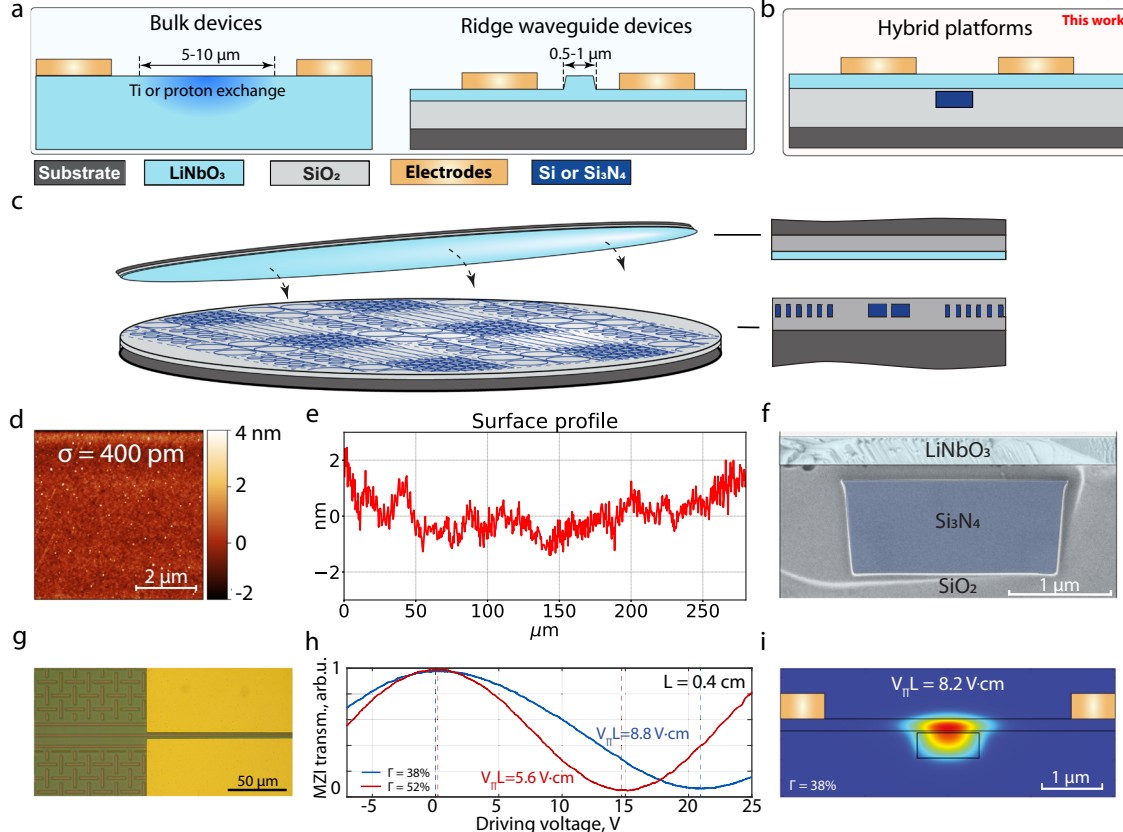

**Fig. 1 | Heterogeneously integrated LiNbO$_3$ photonic circuits based on wafer scale bonding. a** Conventional approaches to LiNbO$_3$ photonics consisting of traditional Ti- or proton-exchange-based waveguides and the recently emerged integrated photonics based on etching of thin-film LNOI (mostly to create ridge waveguides). **b** The hybrid approach presented in this work, involving heterogeneous integration of thin-film LiNbO$_3$ onto Si$_3$N$_4$ wafers with integrated photonic circuits. **c** Schematic showing our approach of bonding a 4 inch (100 mm) thin-film LiNbO$_3$ wafer onto planarized low-loss Si$_3$N$_4$ photonic integrated circuits. **d** AFM image of the Si$_3$N$_4$ Damascene wafer before bonding showing 400 pm RMS roughness over an area 5 μm by 5 μm. **e** Long-range profilometry scan of the Si$_3$N$_4$ Damascene wafer before bonding. **f** False-colored scanning electron micrograph of a cross-section of the hybrid structure. **g** Optical microscope image of a hybrid waveguide with gold electrodes. **h** Half-wave voltage measurements for phase modulators with 4 mm length using a Mach-Zehnder interferometer. **i** FEM simulations of the hybrid optical mode profile for a waveguide with 38% optical mode participation in lithium niobate and 6 μm separation between electrodes.

addition the approach is also extendable to larger wafer sizes (e.g. 6 or 8 inch) and enables further processing using standard fabrication techniques, such as deep-ultraviolet (DUV) lithography used in this work for the adiabatic transitions fabrication and metal lift-off. Given the abundance of $LiNbO_3$ in contrast to III-V materials, widely used in die bonding, and the availability of LNOI in large wafer sizes, the wafer-level integration becomes an attractive and cost effective method for heterogeneous LNOI PICs. Our approach combines the maturity of $Si_3N_4$ integrated photonics with the Pockels effect of $LiNbO_3$ and enables hybrid PICs that exhibit low propagation loss (8.5 dB/m).

## Results

The process flow for our hybrid PICs starts with the fabrication of $Si_3N_4$ waveguide structures using the photonic Damascene process[39,40]. We use a 100 mm-diameter silicon wafer with 4 μm-thick wet thermal $SiO_2$, followed by DUV stepper lithography, preform dry etching, preform reflow, low-pressure chemical vapor deposition (LPCVD) deposition of $Si_3N_4$, chemical-mechanical polishing (CMP), and $SiO_2$ interlayer deposition and annealing, as detailed in the Supplementary Information. The $Si_3N_4$ photonic Damascene process is free of crack formation in the highly tensile LPCVD $Si_3N_4$ film and provides high fabrication yield and ultra-low propagation loss (<3 dB/m). In addition, double-inverse nanotapers[41] are incorporated for efficient edge coupling to lensed fibers. We next perform direct wafer bonding to the wafer with the $Si_3N_4$ photonic integrated circuit. One of the key advantages of the photonic Damascene process is that it includes a surface planarization step. However, direct bonding to the planarized surface is not possible due to long range topography. Therefore, we instead perform CMP on an additional $SiO_2$ interlayer that is deposited and subsequently bond the fabricated $Si_3N_4$ Damascene substrate to a commercially available LNOI wafer (NanoLN). The critical prerequisites for achieving high bonding yield, the surface roughness and topography, are measured prior to bonding. Atomic force microscopy (AFM) measurements over a range of a few microns reveal a root-mean-square (RMS) roughness of 400 pm (Fig. 1e). It is equally important that the long-range topography be reduced to $\mathcal{O}$(nm) over $\mathcal{O}$(100μm). This is achieved with the interlayer silicon oxide that is deposited prior to CMP. The result is long-range topography not exceeding 4 nm over 300 μm, as shown in Fig. 1e - even in the presence of the underlying $Si_3N_4$ photonic integrated circuits. We found this roughness level to be sufficiently low for direct wafer bonding. The donor and the acceptor wafer (the $Si_3N_4$ substrate and the LNOI wafer, respectively) are cleaned, and both are coated by atomic layer deposition (ALD) with a few nanometers of alumina ($Al_2O_3$). The wafers are then bonded and annealed at 250 °C to enhance the bonding strength. After optimization of the CMP treatment, we have successfully bonded several wafers, including whole 100 mm LNOI wafers, with excellent yield - no bonding defects except at the very edge of the wafer, where several areas of a few μm² were debonded. The use of standard cleanroom techniques combined with the high bonding yield makes our processing efficient for the future large-scale LNOI integration with higher throughput compared to the state-of-the-art heterogeneous die-level lithium niobate bonding approaches. A scanning electron microscope image (Fig. 1f) of a cross-section of the layer structure reveals clean bonding results.

To illustrate the versatility, lithographic precision, complexity, and yield of the hybrid platform, we design a reticle with various devices. Figure 2e shows the design layout of the $Si_3N_4$ photonic integrated circuits for a 100 mm wafer; it contains nine fields with 16 chips each – in total, more than 100 chips with dimensions of 5 mm × 5 mm. The reticle includes chips with different types of devices: (1) microresonators with a free spectral range (FSR) of either 100 GHz or 21 GHz, the former being used for electro-optic comb generation; (2) photonic molecules consisting of a pair of coupled

microresonators each with a FSR of 50 GHz, as used for microwave-optical conversion schemes; and (3) waveguides with a length of several centimeters for supercontinuum generation. Finite-element-method simulations (see Fig. 1i) indicate that, for the waveguide and wafer parameters used in this design, the optical mode participation factor for $LiNbO_3$ at the telecommunication wavelength of 1550 nm is $\Gamma = \iint_{LiNbO_3} |E|^2 dS / \iint_{\Omega} |E|^2 dS = 12\%$. For a separate fabrication run we design photonic circuits with straight waveguides and multiple adiabatic transitions between silicon nitride and hybrid optical mode having larger (38–55%) mode participation in lithium niobate due to a different waveguide aspect ratio compared to the previous low-participation design (cf. Methods). To demonstrate the electro-optic capabilities of the heterogeneously integrated $LiNbO_3$ photonic circuits, we deposit either tungsten or gold electrodes on top of the $LiNbO_3$ adjacent to the waveguides with an electrode-electrode gap of 6 μm. Figure 1h shows the performance of a phase modulator, with a length of 4 mm (device image in Fig. 1g) and a confinement of 38%. Measuring the phase shift with the use of a Mach-Zehnder interferometer, we extract a $V_\pi$ value of 22 V, which corresponds to a $V_\pi L$ product of approximately 8.8 V·cm (blue line). By reducing the $Si_3N_4$ waveguide width (and therefore increasing the mode participation in the $LiNbO_3$ layer up to 52%) and decreasing the distance between electrodes down to 5.5 μm, we improve the electro-optic performance even further and achieve a $V_\pi L \simeq 6$ V·cm (red line), however at the expense of optical losses (cf. SI section IV).

To measure the linear optical loss, evanescent coupling properties, and group velocity dispersion (GVD) of the hybrid structures, we perform broadband frequency-comb-assisted spectroscopy[42] of multiple microresonators across the entire wafer with three different external-cavity diode lasers covering the wavelength ranges of 1260–1360 nm, 1355–1505 nm, and 1500–1630 nm. The intrinsic quality factors of individual 100-GHz microring resonators reach up to $Q = 3 \times 10^6$, while the 50-GHz photonic dimers (i.e., coupled microrings) and 21-GHz single rings exhibit even higher quality factors up to $Q = 4.5 \times 10^6$. The latter corresponds to a linear propagation loss of 8.5 dB/m. We observe an absorption peak at approximately 1420 nm (207 THz), which we associate with an overtone of OH-bond vibrations in lithium niobate[43,44]. As shown in Fig. 2g, optical losses rise with increasing optical frequency. We associate this dependency with increased whispering-gallery (radiation) loss as the mode shifts into the $LiNbO_3$ thin film at higher frequencies and becomes less confined (see Fig. 2f). For the same reason, we observe nearly uniform evanescent coupling of optical microresonators over a span of 55 THz. The layout in Fig. 2e is labeled with the most probable linewidth measured for 21-GHz microresonators in various regions, indicating the degree of variation across the wafer (see Supplementary Information). These results not only demonstrate high yield and wafer-scale fabrication but include some of the highest quality factors achieved to date with integrated $LiNbO_3$ devices. Notably, the $Q$s reported here are not isolated values, as in prior work on ridge resonators[9], but are consistently high, i.e., we measure hundreds of resonances with $Q$ above $4 \times 10^6$ (linewidth below 50 MHz), as shown in Fig. 2h. Our hybrid platform also offers the possibility of dispersion engineering; the dispersion can be adjusted by varying the $Si_3N_4$ waveguide geometry or the $LiNbO_3$ thickness. In this work, we designed the structure to work in the near-zero GVD regime advantageous for broadband optical frequency comb generation. As shown in Fig. 2d for a microresonator with a FSR of 100 GHz, the measured integrated microresonator dispersion ($D_{int}$) only varies by 15 GHz over an optical bandwidth of 55 THz. The presented design demonstrates an example of devices that are uniformly coupled over a broad frequency range and, at the same time, experience flat integrated dispersion with $D_2/2\pi$ of $\mathcal{O}$(100 kHz). Note that flat and anomalous dispersion can be achieved for the waveguide configuration with low participation in $LiNbO_3$ (950 nm waveguide thickness), while for the electro-optic configuration (600 nm waveguide

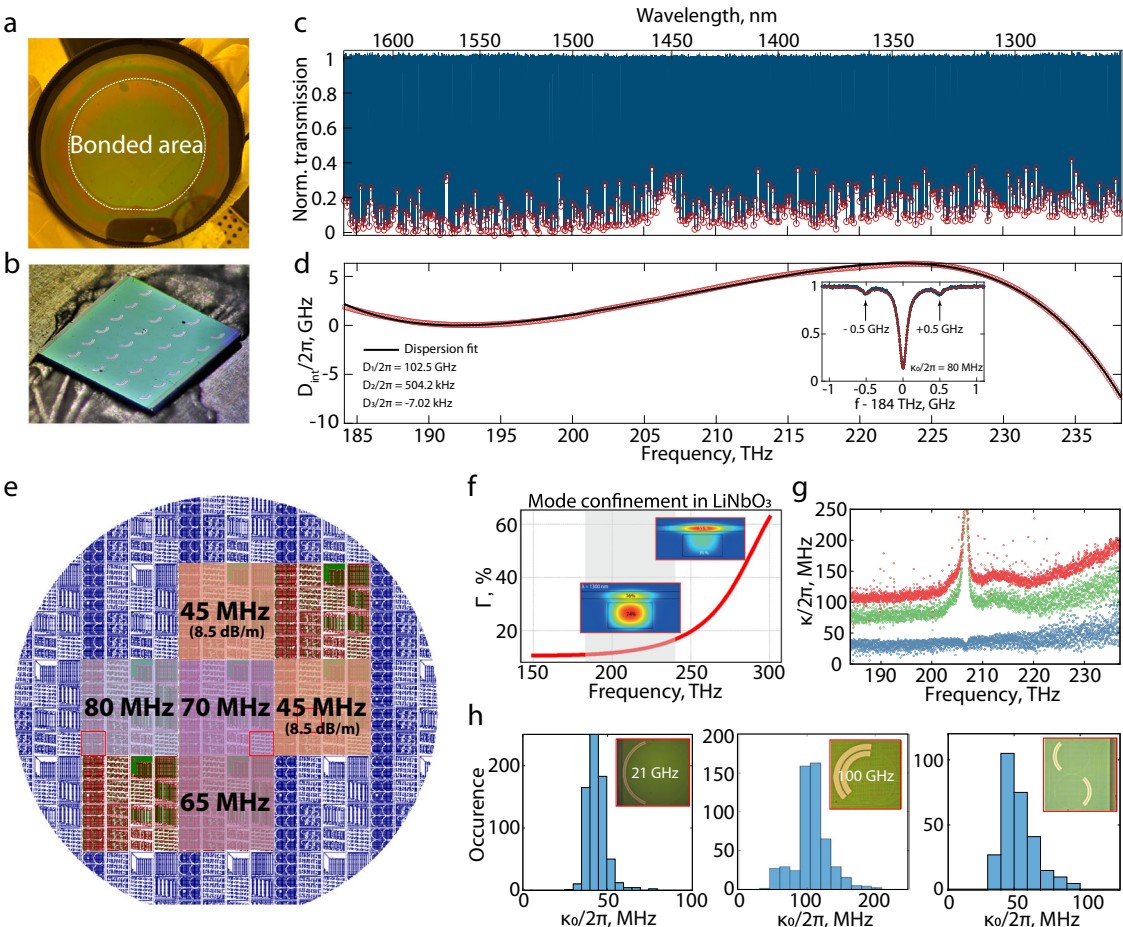

**Fig. 2 | Optical loss measurements. a** Photograph of a 3-inch LiNbO₃ wafer fully bonded to a 4-inch photonic Si₃N₄ Damascene wafer. **b** Photograph of an integrated photonic chip (5 mm × 5 mm size) with electrodes. **c** Broadband transmission measurements of a ring resonator with a FSR of 100 GHz showing flat coupling of resonances for wavelengths from 1260 nm to 1630 nm. **d** Extracted integrated dispersion of the ring resonator. The inset shows an expanded view of one of the modes at 184 THz with amplitude modulation sidebands at 500 MHz and the corresponding fitting curve. **e** Wafer map with average linewidth indicated for the ring resonators with a FSR of 21 GHz used as a reference. **f** Simulated optical mode confinement in LiNbO₃ as a function of optical frequency. Insets show typical mode profiles. The grey-shaded area represents the range measured. **g** Loaded (red), intrinsic (green), and external coupling (blue) linewidth of a 21-GHz microring (F6, see Supplementary Informaton). **h** The measured linewidth of three types of devices on the wafer accumulated over a measurement bandwidth of 55 THz.

thickness) the dispersion is strongly normal for any waveguide width. The trade-off between dispersion engineering and electro-optic interaction strength is a question of specific designs for specific experiments.

Efficient input coupling from an optical fiber to the photonic chip is paramount for numerous applications. For air-cladded LiNbO₃ ridge waveguides, inverse tapers lead to fiber-chip edge-coupling losses of 5–10 dB per facet unless complicated multi-layer etching is used[27,45,46]. This is due to the significant mode mismatch between the lensed fiber mode (typically circular with about 2.5 μm diameter) and the asymmetric mode of partially etched air-cladded ridge waveguide structures. Some recent work on integrated LiNbO₃ devices demonstrated the possibility of using embedded silicon edge-couplers to overcome this challenge[17]. In our case, the geometrical mode profile mismatch would lead to significant coupling loss if the LiNbO₃ layer remained on top of underlying Si₃N₄ inverse tapers. Hence, we remove the LiNbO₃ from the coupling regions and rely on standard Damascene Si₃N₄ inverse tapers[41]. While this provides an efficient input coupling, there remains the challenge of the transition between the regions with and without LiNbO₃. To address this issue, we designed and implemented adiabatic tapers in the LiNbO₃ layer, as shown in Fig. 3c. The inverse tapers are 100 μm long with a tip width of 500 nm and a final width of 10 μm. Both film removal and etching of the tapers are done in a single

fabrication step using argon ion-beam etching with a photoresist etch mask. Crucially, all the functional photonic components do not depend on the etching of the LiNbO₃ tapers, which are employed only at mode transition regions and which, due to the short taper length, does not require low roughness. We thus keep the LiNbO₃ layer unprocessed for all the photonic components, where both roughness and precise alignment are critical to achieving low optical losses.

To measure the efficiency of the adiabatic transitions and remove the ambiguity associated with the fiber-chip coupling loss from the measurement, we designed an experiment in which we introduce multiple breakouts on straight waveguides, where the optical mode experiences transitions from Si₃N₄ waveguides into the hybrid Si₃N₄-LiNbO₃ mode and back, as depicted in Fig. 3b. We fabricated waveguides with two, four, six, and 10 transitions (input/output tapers and zero, one, two, and four breakouts, respectively) to determine the increase in loss due to each transition. Figure 3d shows the results of these measurements. As a reference, we compare them with straight interfaces (schematic in Fig. 3e). As can be seen in Fig. 3f, each straight transition leads to approximately 1 dB loss, whereas the tapered input/output behaves in this measurement as a virtually lossless transition. For the case of the tapered transitions, we observe approximately 0.8 dB additional loss for ten interfaces, as shown in Fig. 3d. Considering the statistical uncertainty in the measurements, we deduce a

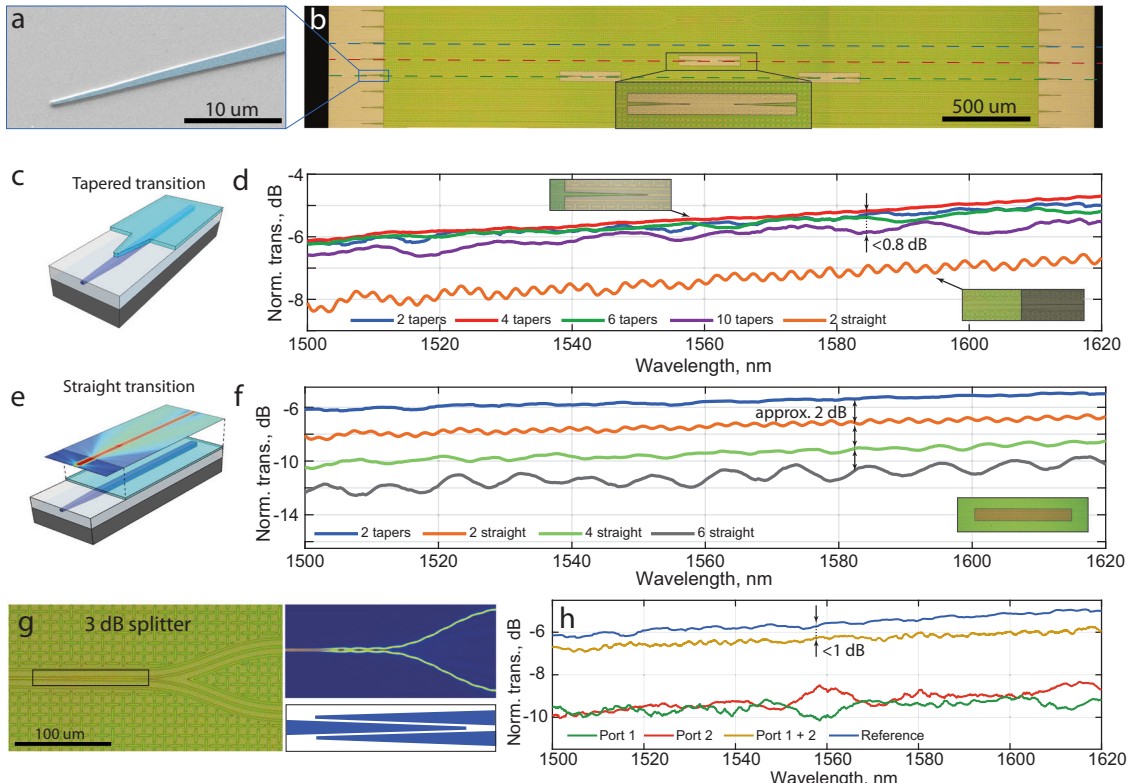

**Fig. 3 | Adiabatic mode transitions and optical splitters. a** False-colored SEM image of a LiNbO$_3$ taper fabricated to make a low loss optical mode transition from a purely Si$_3$N$_4$ waveguide to the hybrid Si$_3$N$_4$-LiNbO$_3$ waveguide. **b** Optical microscope image of a chip used to estimate the performance of interface tapers. Horizontal dotted lines mark waveguides with two (blue), four (red), six (green), and ten interface transitions (purple). Ten transitions induce only 0.8 dB total loss. The inset shows an expanded view of a breakout region. **c** Schematic of a tapered (adiabatic) interface transition. **d** Transmission measurements of waveguides with breakouts as described in **b** as well as a typical straight (non-adiabatic) transition for comparison (orange line). **e** Schematic of a straight (non-adiabatic) interface transition. **f** Transmission measurements of waveguides with straight transition breakouts. **g** Optical image, simulation, and schematic of a W-type 3 dB splitter. The geometry is defined by the underlying Si$_3$N$_4$ layer. **h** Transmission measurements of two ports of the splitter together with the transmission of a straight waveguide as reference.

transition loss of <0.1 dB per taper. Note that the presented results are achieved with the high-participation waveguide configuration (600 nm Si$_3$N$_4$ thickness).

The lithographic precision of the Si$_3$N$_4$ photonic circuit layer provides our heterogeneous integration approach with versatiliy and robustness, as confirmed by the implementation of a W-shaped 3 dB splitter/coupler[47] (see Fig. 3g) that uses the hybrid Si$_3$N$_4$-LiNbO$_3$ mode but is defined solely by underlying Si$_3$N$_4$ inverse tapers. Splitters are important components for many optical devices, such as electro-optic modulators, optical networks, and lasers based on reflective semiconductor optical amplifiers. The elegance of this type of splitter is in its simplicity of design. Due to the presence of the LiNbO$_3$ slab and the single-mode nature of our hybrid waveguides, the optical mode is adiabatically transferred from the input arm to the output arms. We make the tapered sections 100 µm long, ensuring a small footprint for integrated components exploiting this design. Transmission measurements of the device reveal a flat response, with power asymmetry between the two arms not exceeding 1.7 dB and on-chip insertion loss not exceeding 1 dB in the 1500–1620 nm wavelength range (Fig. 3h).

To demonstrate the electro-optic performance achievable with the hybrid LiNbO$_3$ microresonators, we generate electro-optic frequency combs in microresonators with a FSR of 21 GHz pumped resonantly in the telecommunications C-band. We apply a high-power (40 dBm) microwave signal with a frequency of 20.97 GHz across the integrated electrodes such that microwave-induced sidebands are resonantly enhanced (Fig. 4a, b). As only 12% of the optical mode is confined inside the lithium niobate, the phase modulation amplitude

at 40 dBm injected microwave power is approximately $0.14\pi$ (see Supplementary Information section VII). The electro-optic coupling is enhanced due to the device's high-quality factor and flat dispersion. We observe around 60 sidebands within a 25 dB span, as depicted in Fig. 4c. We also make use of the previously mentioned homogeneous coupling of our hybrid devices at optical wavelengths ranging from 1260 nm to 1630 nm to generate electro-optic combs at five different pump wavelengths (1290 nm, 1345 nm, 1500 nm, 1550 nm, and 1625 nm) with a single device (Fig. 4d).

According to the simulations presented in Supplementary Information section IV and the measurements of 4 mm-long phase modulators with 38% mode participation in LiNbO$_3$, the geometry can be optimized for maximum electro-optic efficiency with a characteristic $V_\pi \cdot L$ product comparable to (2x larger than) the performance of X-cut ridge-waveguide platforms[16] with similar electrode-induced loss. In ridge waveguides, most of the electro-optic interaction occurs not in the ridge itself, but in the slab layer, where the modulating electric field is an order of magnitude stronger. Therefore, the ridge waveguides and hybrid waveguides are conceptually similar in terms of electro-optic interactions (i.e., the role of the slab in ridge waveguides is being taken over by the bonded LiNbO$_3$ layer in our hybrid structure). Similar studies on optimization of electro-optic performance of heterogeneously integrated LiNbO$_3$ devices can also be found elsewhere[32]. As a further example of the electro-optic capabilities, we fabricate photonic dimers, which are known building blocks for quantum-coherent transducers based on cavity electro-optics[19,48,49]. Figure 4g shows the mode hybridization in the system as a function of the applied DC

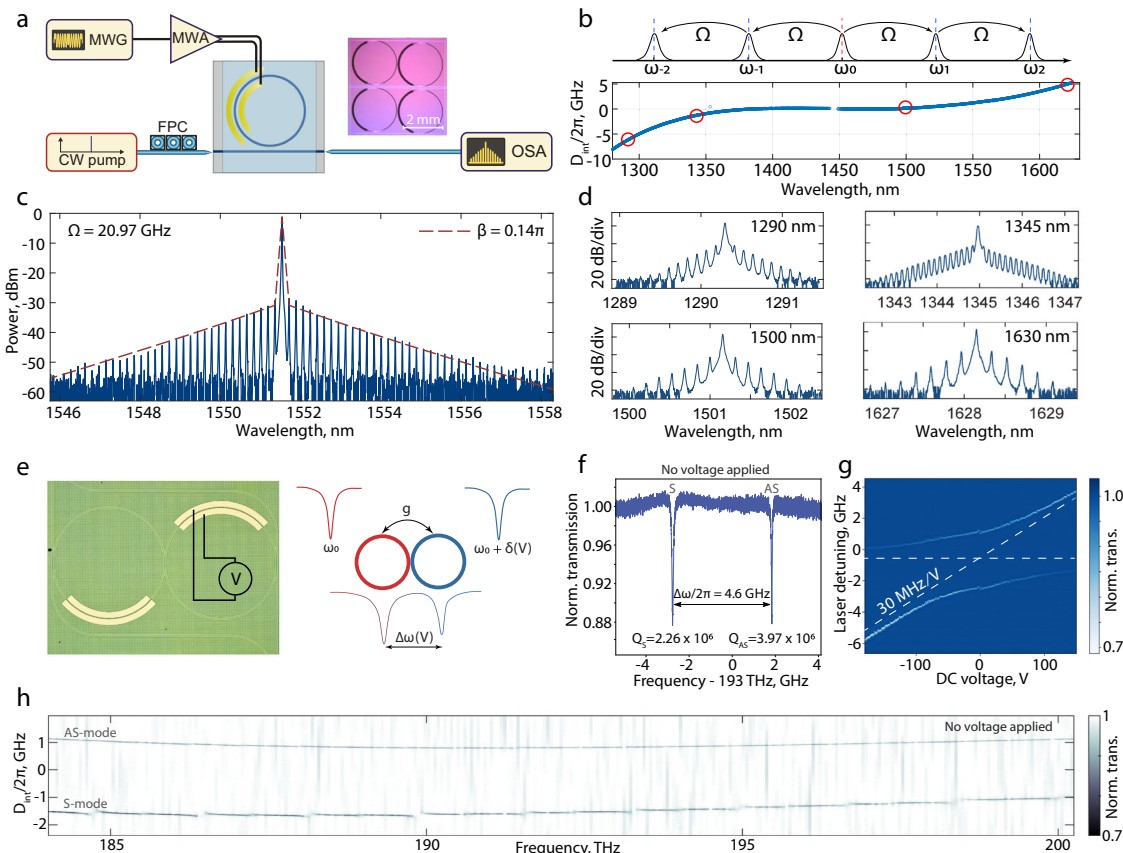

**Fig. 4 | Electro-optic frequency comb generation and tunable photonic dimers.** **a** Experimental setup for electro-optic frequency comb generation. MWG - microwave generator, MWA - microwave amplifier, OSA - optical spectrum analyzer. **b** Mode coupling schematic and integrated dispersion of the device. **c** Measured optical spectrum of the generated electro-optic comb with a central frequency of 1552 nm. The dashed line corresponds to numerical simulations with phase modulation amplitude $\beta = 0.14\pi$. **d** Examples of electro-optic frequency combs generated at four other pump wavelengths. **e** Photonic dimer image and illustration of mode hybridization. **f** Splitting of high-Q resonances in a photonic dimer without additional biasing. S - symmetric supermode, AS - antisymmetric supermode. **g** DC tuning of the photonic dimer mode hybridization, corresponding to linear tuning of around 30 MHz/V for a single mode. **h** Echelle-type spectrogram of the photonic dimer transmission, showing mode hybridization over a broad scanning range.

voltage. We observe frequency tuning of 30 MHz/V when a DC voltage is applied to one of the rings. Moreover, the precise and mature fabrication of the $Si_3N_4$ waveguides enables the creation of high-Q photonic dimers exhibiting broadband normal mode splitting even at zero-bias (see Fig. 4f–h). The presence of avoided mode crossings for the symmetric supermode (lower frequency) is due to interaction with higher order modes, which is common with the photonic dimer configuration[50]. In one last experiment exploiting the $\chi^{(2)}$ nonlinearity of the $LiNbO_3$, we perform supercontinuum generation in the hybrid waveguides. We observe octave-spanning supercontinuum generation mediated by the $\chi^{(3)}$ nonlinearity, together with simultaneous second-harmonic generation due to the optical field in the $LiNbO_3$, allowing direct measurement of the carrier-envelope offset frequency of the femtosecond pulse laser used as a pump. The details of this experiment can be found in the Supplementary Information section VIII.

To conclude, we have demonstrated a hybrid $Si_3N_4$-$LiNbO_3$ platform for photonic integrated circuits using direct wafer-scale bonding that endows the mature low-loss $Si_3N_4$ technology with the second-order nonlinearity ($\chi^{(2)}$ / Pockels effect) of $LiNbO_3$. The heterogeneous integration preserves the precise lithographic control, low propagation loss, and efficient fiber-to-chip coupling of the underlying $Si_3N_4$ waveguides for use in a variety of important photonic building blocks. We have also presented a design for the transition from $Si_3N_4$ waveguides to hybrid $Si_3N_4$- $LiNbO_3$ waveguides with a measured insertion loss not exceeding 0.1 dB per interface. The ability to achieve low-loss transitions is essential for the realization of complex devices,

providing a bridge between passive silicon nitride photonics and electro-optic devices. We achieve a $V_\pi L$ product value of approximately 8.8 V · cm for the phase modulators (single-arm) in an electrode configuration with negligible induced loss. To the best of our knowledge, this is the first time a heterogeneously integrated $LiNbO_3$ photonic platform combines all the beneficial features of $Si_3N_4$ PICs at wafer scale. A comparison of the simultaneously achieved desirable features is given in Supplementary Table 1. The electro-optic performance depends on the optical mode confinement in the $LiNbO_3$ slab layer and can reach levels comparable to that of ridge waveguide structures while keeping propagation losses independent of the quality of the $LiNbO_3$ etching. Possible applications of our platform include photonic switching networks for neuromorphic or quantum computing, devices for quantum state transduction from microwave to optical photons, integrated electro-optic frequency comb sources, on-chip generation of second-harmonic and squeezed light, as well as high-speed electro-optic devices for optical communications or rapidly tunable, low-noise lasers.

## Methods
### Fabrication
The schematics for all the fabrication steps taken are presented in Supplementary Figure 1. After the first CMP process, the surface is not yet ready for wafer bonding: In order to clean the silica particles from the CMP slurry, a short dip in buffered hydrofluoric acid (BHF) is performed. This generates some topography between the silicon

nitride structures and the surrounding silicon oxide cladding. To obtain a bonding-ready surface, an interlayer of a few hundred nanometers of silicon oxide is deposited by LPCVD and subsequently polished down and planarized. As described in the main text, after this step, the roughness and long-range non-uniformity are low enough for bonding. The remaining interlayer thickness is about 100 nm. Moreover, reflectometry measurements revealed that the thickness variation of the interlayer was about 5 nm over the entire wafer. In order to bond the LNOI wafer, the surface of both wafers is cleaned, and a few nanometers alumina are deposited by ALD on both of them. The wafers are then brought into contact. To increase the bonding strength, the bonded wafers are annealed for several hours at 250 °C. After bonding, the Si of the LNOI carrier is removed by grinding and tetramethylammonium hydroxide. The buried oxide of the LNOI carrier is removed with BHF. Tungsten is then sputtered onto the bonded lithium niobate and patterned into electrodes with fluorine-based reactive ion etching (RIE). The next step is to pattern the bonded lithium niobate thin film. The objective here is two-fold: remove the $LiNbO_3$ from the chip facets and fabricate tapers to ensure a smooth transition in the waveguide as discussed in the main text. To this end, a $SiO_2$ protection layer is first sputtered on the $LiNbO_3$ surface. Ion beam etching is then employed with a photoresist etch mask to etch both the $SiO_2$ protective layer and the $LiNbO_3$, leaving only a slab of less than 50 nm thickness. The etching is performed with an argon ion beam impinging at an angle of 30° while the wafer is rotating. In order to reduce the accumulation of sputtered material during etching, the photoresist mask is reflown at 130 °C for 3'30" after exposure and development. After etching and removal of the photoresist, the sputtered material is removed by wet etching in a concentrated solution of ammonia and hydrogen peroxide heated to 85 °C. During this cleaning, the surface of the $LiNbO_3$ thin film is protected by the $SiO_2$ layer. After cleaning, the $SiO_2$ layer can be removed using a BHF solution. In this process step, the thin remaining $LiNbO_3$ slab acts as a protective layer to prevent the BHF from etching the interlayer covering the $Si_3N_4$ waveguides and undercutting the $LiNbO_3$ tapers. To release the chips, the edges are defined by dry-etching the $SiO_2$ cladding with fluorine-based chemistry, and then further down into the Si carrier using the Bosch process. This ensures a relatively smooth chip facet that does not require further polishing. The chips are then separated by grinding the wafer down to the etched depth from the backside.

**Waveguide geometry**
The X-cut LNOI wafer used in the work is provided by NanoLN, with a $LiNbO_3$ thickness of 300 nm. We use two configurations of optical waveguides: 950 nm-thick $Si_3N_4$ (wafer D67b01, low-participation regime) and 600 nm-thick $Si_3N_4$ (wafer D8812, high-participation regime), and keep an interlayer $SiO_2$ layer of about 100 nm thickness. The waveguide width ranges from 1.0–2.0 µm.

**Transmission calibration**
The absolute fiber-to-fiber transmission measurements are done in two steps as depicted in Supplementary Fig. 3. In the first step we couple the light into the chip with lensed fibers and record the transmission using a photodetector. To eliminate any uncertainty in input laser power we split the signal and track it directly, using an additional photodiode. Then we connect the input and output arms of the setup directly using a fiber patch cord. By recording the same metrics for a directly connected patch cord we are able to normalize the photonic chip transmission measurements.

## Data availability
Data used in this study are available from the corresponding author upon request. Data used to produce the plots within this paper is available on Zenodo at https://doi.org/10.5281/zenodo.7781982.

## Code availability
Code used in this study are available from the corresponding author upon request. Code used for the EO comb simulations is available on GitHub at github.com/ElKosto/PyCORe.

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

## Acknowledgements

This work was supported by funding from the European Union Horizon 2020 Research and Innovation Program under the Marie Sklodowska-Curie grant agreement No. 722923 (OMT) and No. 812818 (MICRO-COMB), as well as under the FET-Proactive grant agreement No. 732894 (HOT). This work was also supported by the Swiss National Science Foundation under grant agreement No. 176563 (BRIDGE) and 186364 (QuantEOM), as well as by Contract HR0011-20-2-0046 (NOVEL) from the Defense Advanced Research Projects Agency (DARPA), Microsystems Technology Office (MTO). We thank the Operations Team of the Binnig and Rohrer Nanotechnology Center (BRNC), especially Diana Davila Pineda and Ronald Grundbacher, for their help and support. Silicon nitride substrates were fabricated in the EPFL center of Micro-NanoTechnology (CMi). J.R. acknowledges support from the SNSF via grant number 201923 (Ambizione). We also thank Aleksandr Tusnin for his help in numerical simulations.

## Author contributions

P.S. and T.J.K. initiated the study and supervised the project; M.C. developed the idea and performed numerical analysis and design; A.R., R.N.W., C.M. and J.L. developed the processes and fabricated the samples with the assistance from S.H.; D.C. bonded the wafers; U.D., Y.P. and A.R. performed CMP; R.N.W. and V.S. developed adiabatic tapers etching procedure; M.C., T.B., and M.A.A. performed the experiments and data analysis with the assistance of A.S., and J.R; M.C., R.N.W. and A.R. wrote the manuscript with contributions from all the authors.

## Competing interests

T.J.K. is a co-founder and shareholder of LiGenTec SA and Luxtelligence SA, start-up companies engaged in making $Si_3N_4$ and $LiNbO_3$ photonic chips available via foundry service. Other authors declare no competing interests.
