## [Peer review file · Nature Communications]

REVIEWER COMMENTS

Reviewer #1 (Remarks to the Author):

I appreciate the efforts of the authors in improving the transition losses by fabricating the etched tapers in lithium niobate. However, based on the authors' replies, I believe they might have misunderstood the main concerns from my earlier report, namely the low conceptual novelty with respect to state of the art, the rationale of the approach, and the limited performance of the modulators.

1. While it is clear that this is the first time that a LN wafer is bonded on silicon nitride photonic integrated circuits, it remains unclear whether this approach provides important advantages with respect to die bonding (earlier works) given that the latter: a) also enables integration at the wafer level; b) is easier to implement as it has relaxed requirements in terms of wafer homogeneity; c) optimizes the real state of an expensive wafer. The tradeoffs between these approaches should be better conveyed and acknowledged in the introduction.

2. Regrettably, my earlier comment "The electro-optic modulator has an extremely large $V_{\pi-L}$. This value is not only significantly higher than ridge LN waveguides (which is expected) but also higher than the results presented in ref. 45 (6.67 V-cm) that relies on a hybrid waveguide that combines bonding of LN on a processed silicon nitride circuit. The authors argue that the $V_{\pi-L}$ can be improved with an optimized design of the core (cf. Fig. 3 in S3), but this results into a fundamental tradeoff with optical field confinement and, I suspect, the ability to achieve broadband dispersion engineering of the hybrid mode which is a key property enabled by the thick Si₃N₄ waveguides fabricated with the Damascene process." is not well addressed.

The authors do acknowledge that their modulators display lower performance, and they mention in their reply that "The critical aspect of our work is not the electro-optic efficiency itself but the combination of low optical losses at wafer-scale (< 0.1 dB/cm and 2.5 dB/facet insertion loss) with electro-optics ($V_{\pi-L} \approx 30$ Vcm) that can be optimized down to approximately 8 Vcm without the need to etch lithium niobate, therefore preserving the low optical losses". However, it remains unclear whether the optimization process (which essentially relies on modifying the aspect ratio of the silicon nitride waveguide to enhance the confinement in the LN slab) can indeed preserve the low optical losses, keep the transition losses low and attain dispersion engineering.

Without further compelling evidence of the importance of wafer-level manufacturing and a more exhaustive analysis of the fundamental tradeoffs in optical confinement and loss when doing the optimization process of the silicon nitride waveguide, I cannot recommend the publication of the work in Nature Communications.

Reviewer #2 (Remarks to the Author):

I appreciate that the authors responded in detail to my previous comments. The paper is reorganized and has toned down many of its original claims. The authors, for the most part in the rebuttal, acknowledged that my original three major concerns are valid: 1) The novelty of the platform; 2) CMOS compatibility; 3) The performance. The question remains – what makes this paper stand out compared to the references 47 (Boynton et al., Optics Express 2020), and 48 (Chang et al, Optics Letter 2017)? I agree with the authors that, as a platform, this work demonstrates potential for building complex circuits using wafer-scale PDKs. There are advances at various levels in each demonstrated building block. However, these blocks remain components and do not enable complex functional circuits.

A point-by-point response to the reviews of the manuscript “A heterogeneously integrated lithium niobate-on-silicon nitride photonic platform”

First of all, we thank the reviewers for their attentive reading of the new version of our manuscript. We appreciate the comments given by both referees and we give an explicit reply in this letter. Following the reviewers' questions, we made several changes to the manuscript that we believe will help to better convey the novelty of our work. For convenience, the referees' comments are colored black while our responses use blue, and changes to the manuscript are colored red both here and in the manuscript file.

Referee #1:

I appreciate the efforts of the authors in improving the transition losses by fabricating the etched tapers in lithium niobate. However, based on the authors' replies, I believe they might have misunderstood the main concerns from my earlier report, namely the low conceptual novelty with respect to the state of the art, the rationale of the approach, and the limited performance of the modulators.

We thank Reviewer #1 for his/her positive feedback on the transition losses. We also agree that the question of novelty should be addressed properly, as in the previous reply letter we focused more on the reviewer's technical remarks. In this reply letter, we will focus only on the novelty issues raised by the reviewer and we hope we will be able to meet his/her expectations.

1. While it is clear that this is the first time that a LN wafer is bonded on silicon nitride photonic integrated circuits, it remains unclear whether this approach provides important advantages with respect to die bonding (earlier works) given that the latter: a) also enables integration at the wafer level; b) is easier to implement as it has relaxed requirements in terms of wafer homogeneity; c) optimizes the real state of an expensive wafer. The tradeoffs between these approaches should be better conveyed and acknowledged in the introduction.

The question of the advantages and disadvantages of the wafer-level bonding approach presented in our work is indeed an important one to be discussed in the manuscript. Die-level bonding can be advantageous in some cases, as it was stated by the Reviewer. We cite here a related work from the field of MEMS and semiconductor processing, discussing the question of choice between wafer-wafer stacking and chip-wafer/chip-chip stacking integration:

T. Matthias et al., “3D Process Integration – Wafer-to-Wafer and Chip-to-Wafer Bonding”, MRS Online Proceedings Library, **970**, 408 (2006)

“Wafer-level integration has the advantage of higher throughput, enhanced cleanliness and the flexibility that standard fab equipment can be used for further processing. 3D integration applying chip-to-wafer bonding focuses on the yield (“good known die”) and enables to stack dies of different size e.g. several small dies on one big base die.”

In other words, the wafer-level bonding is not preferable in cases, where the bonding yield is not good enough to risk the whole wafer, or if the acceptor wafer needs complicated processing. In these cases, as it was stated by the Reviewer, the die-level approach indeed “optimizes the real state of an expensive wafer”. However, in our case of LNOI wafer bonding, we treat the acceptor wafer with **standard cleanroom techniques** (CMP, ALD, etc., see SI section I for details). Moreover, the **bonding yield** of our process is excellent with micrometer-scale de-bonded areas at the edges of the wafer as we state in the manuscript. This is also proved by the fact that the devices we measure across the wafer perform the same way.

We also cite this paper in our revised version of the manuscript and put the following sentences in the introduction:

Compared to die-level integration, wafer-scale bonding has several distinct advantages, chief among them a significantly increased wafer throughput that is advantageous in high volume applications [52]. In addition the approach is also extendable to larger wafer size (e.g. 6 or 8 inch) and enables further processing using standard fabrication techniques, such as DUV lithography used in this work for the adiabatic transitions fabrication and metal lift-off. Given the abundance of LiNbO_3 in contrast to III-V materials, widely used in die bonding, and the availability of LNOI in large wafer sizes, the wafer-level integration becomes an attractive and cost effective method for heterogeneous LNOI PICs.

2. Regrettably, my earlier comment “The electro-optic modulator has an extremely large $V_{\pi L}$. This value is not only significantly higher than ridge LN waveguides (which is expected) but also higher than the results presented in ref. 45 (6.67 V-cm) that relies on a hybrid waveguide that combines bonding of LN on a processed silicon nitride circuit. The authors argue that the $V_{\pi L}$ can be improved with an optimized design of the core (cf. Fig. 3 in S3), but this results into a fundamental tradeoff with optical field confinement and, I suspect, the ability to achieve broadband dispersion engineering of the hybrid mode which is a key property enabled by the thick Si_3N_4 waveguides fabricated with the Damascene process.” is not well addressed.

The authors do acknowledge that their modulators display lower performance, and they mention in their reply that “The critical aspect of our work is not the electro-optic efficiency itself but the combination of low optical losses at wafer-scale (< 0.1 dB/cm and 2.5 dB/facet insertion loss) with electro-optics ($V_{\pi L} \approx 30$ Vcm) that can be optimized down to approximately 8 Vcm without the need to etch lithium niobate, therefore preserving the low optical losses”. However, it remains unclear whether the optimization process (which essentially relies on modifying the aspect ratio of the silicon nitride waveguide to enhance the confinement in the LN slab) can indeed preserve the low optical losses, keep the transition losses low and attain dispersion engineering.

We apologize for being not so clear in our previous reply, but we draw the reviewer’s attention to the fact that we demonstrate phase modulators having 8.8 Vcm and 5.6 Vcm $V_{\pi L}$ values. As stated in the text of the manuscript, we observe insertion loss increase for the highest participation case (52%), while for the 8.8 Vcm configuration we see no additional loss due to the close proximity of electrodes to the optical waveguide. Therefore, we indicate 8.8 Vcm as our main $V_{\pi L}$ value metric. Those values are not simulations, but measurements (which correspond well to the

simulations). It appears that in the previous reply letter, we did not update the $V_{\pi}L$ metric in the comparison table. We put the latest version of the comparison table of the manuscript (Supplementary Table 1) here:

Reference	Intrinsic Q-factors	Linear optical loss	$V_{\pi}L$ product (Tuning rate)	Insertion loss	Wafer-level fabrication	Statistical analysis
This work	$4 \cdot 10^6$	0.1 dB/cm	8.8 V·cm (42 MHz·V ⁻¹)	2.5 dB/facet	Yes	Yes
[13]	10^7	0.027 dB/cm	No data	No data	Yes	No
[14]	No data	No data	13.4 V·cm	6.5 dB/facet	No	No
[15]	$2.5 \cdot 10^6$	No data	(500 MHz·V ⁻¹)	No data	Yes	No
[16]	No data	0.2 ± 0.4 dB/cm	No data	5 dB/facet	No	No
[17]	$1.8 \cdot 10^6$	0.27 dB/cm	No data	1.7 dB/facet ^b	Yes	Yes
[18]	$7.68 \cdot 10^5$	0.2 dB/cm	5.1 V·cm ^a	6.5 dB/facet	Yes	No
[19]	No data	No data	6.2 V·cm	No data	Yes	No
[20]	No data	7 dB/cm	6 V·cm	>10 dB/facet	Yes	No

We also modify Figure 1 panel (h) to demonstrate the measured $V_{\pi}L$ metric explicitly to avoid any confusion for the electro-optic performance:

Figure 1: ... (h) Half-wave voltage measurements for phase modulators with 4 mm length using Mach-Zehnder interferometer. (i) FEM simulations of hybrid optical mode profile for a waveguide with 38% optical mode participation in lithium niobate and 6 μm separation gap between electrodes (see SI for electro-optic simulations).

As indicated in the manuscript text, these measurements are done on the “optimized” silicon nitride waveguides on a separate design run with 600 nm waveguide thickness.

The question of additional optical losses, transition loss, and dispersion engineering of the “optimized” structure is indeed important to be addressed further.

- 1) On the **linear optical loss**. In the last version of our manuscript, we added simulation results of the bending loss for high-confinement waveguide configurations (SI section V). Unfortunately, on the electro-optic design run (D88_12), we have much less optical loss data, as the main aim of the design was to achieve efficient electro-optic modulation, in contrast to the low-confinement design (D67b_01), where we explicitly study optical losses on different microresonator configurations. In the revised version of the manuscript we add linewidth measurements of 2 different microresonators in the “optimized” waveguide configuration with 2 different bending radii.

To verify the model, we measure Q-factors of two microresonators in the electro-optic waveguide configuration: 120 μm and 180 μm radii (Figure S5(c-d)). The spacer thickness is around 80 nm according to the cross-section SEM measurements, and waveguide width is set to be 2 μm for both resonators. The smaller ring reveals an increase in the resonance linewidth, which leads to the Q-factor of $3.9 \cdot 10^5$, while the larger ring has almost a magnitude higher Q-factor ($2.2 \cdot 10^6$), close to the ultimate measured values presented in the main manuscript. The latter means that for 80 nm spacer thickness the loss is not dominated by bending already at 180 μm radius. The “waving” background in panel (c) corresponds to the standing waves inside the bus waveguide and are visible due to the 16-GHz scale of the frequency axis (the total resonance linewidth in this case reaches approximately 800 MHz). We also mark these measurement results with red crosses on the chart presented in Figure S5(a).

Supplementary Figure 5: Bending loss-limited Q-factors in hybrid waveguides. (a) Simulation results for the “electro-optic” waveguide geometry for different bending radii and spacer thicknesses. Crosses correspond to the measurements presented in panels (c-d). (b) Example of an optical mode dissipation in case of a thick spacer. The inner waveguide colors display linear-scale mode distribution, while the far-eld lines are in log scale to visualize an optical mode dissipation in the tail, being de-coupled from the Si_3N_4 waveguide core. (c) Resonance fitting (red curve) of a 120 μm microresonator in the simulated configuration with 80 nm spacer thickness. (d) The same measurement for a resonator with 180 μm radius. Note the frequency scale change for panels (c)-(d).

- 2) On the **transition loss** (fiber-to-chip coupling). The transition loss results we report in the last version of our manuscript, as well as all the measurements displayed in Figure 3, are measured on the high-confinement configuration of the waveguide. In other words, those are results from wafer D88_12. In the revised text we explicitly mention this fact to give our readers a better understanding of which configuration was used in the experiment.

Note that the presented results are achieved on the high-confinement waveguide configuration (600 nm Si₃N₄ thickness).

Please also note that the fiber-to-fiber transmission measurements display not only the transition loss but also include the linear loss in the 5 mm long waveguides in the electro-optically “optimized” aspect ratio configuration. The total insertion loss does not exceed 5 dB fiber-to-fiber, most of which comes from the fiber-to-Si₃N₄ waveguide coupling as we discuss in the manuscript. However, as we stated before, we find the linear loss study based on the fiber-to-fiber transmission measurements not reliable and not accurate enough for the PIC characterization, so we did not include these discussions in the manuscript to not confuse our readers.

- 3) On the dispersion engineering. We thank Reviewer 1 for this question as our initial version lacked the discussion on dispersion engineering for the high-confinement waveguides. If we consider dispersion engineering as the ability to invert dispersion from normal to anomalous by changing the waveguide width, the “optimized” structure indeed lacks it. In the high-confinement regime (waveguide thickness 600 nm) we do see only normal dispersion for the silicon nitride waveguide width range of 0.8-2.0 μm. To make this fact clear for the readers interested in this aspect, we add the following sentence in the revised version next to the dispersion engineering discussions:

The presented design demonstrates an example of devices that are uniformly coupled over a broad frequency range and, at the same time, experience flat integrated dispersion with $D^2/2\pi$ of $O(100 \text{ kHz})$. Note, that the flat and anomalous dispersion can be achieved for the low-confinement waveguide configuration (950 nm waveguide thickness), while for the electro-optic configuration (600 nm waveguide thickness) the dispersion is strongly normal for any waveguide width. The trade-off between dispersion engineering and electro-optic interaction strength is a question of specific designs for specific experiments.

Despite the fact, that the near-zero dispersion control is achieved only for the low-confinement regime, we believe that our platform still remains versatile for many applications, as one can choose between the LN confinement factor and the ability to achieve anomalous dispersion depending on the specific experiment.

Reviewer #2 (Remarks to the Author):

I appreciate that the authors responded in detail to my previous comments. The

paper is reorganized and has toned down many of its original claims. The authors, for the most part in the rebuttal, acknowledged that my original three major concerns are valid: 1) The novelty of the platform; 2) CMOS compatibility; 3) The performance.

We thank Reviewer #2 for his/her positive feedback on the text improvements.

The question remains – what makes this paper stand out compared to the references 47 (Boynton et al., Optics Express 2020), and 48 (Chang et al, Optics Letter 2017)? I agree with the authors that, as a platform, this work demonstrates potential for building complex circuits using wafer-scale PDKs. There are advances at various levels in each demonstrated building block. However, these blocks remain components and do not enable complex functional circuits.

As we state in the response to Reviewer #1, the reliable wafer-scale bonding approach helps to increase the **process throughput**, which is one of the key metrics for the future widespread of integrated lithium niobate photonics. We also improve other metrics compared to the references cited by the Referee, mainly **the optical loss**. We also demonstrate the performance of our platform in **various applications**, which was not shown for heterogeneously integrated LNOI approaches before. As we discussed in the previous reply letter, we believe that the demonstration of complex functional circuits sets the aim for future works, aimed at specific experiments. In the current manuscript, we demonstrate the basic performance of our platform with an extensive analysis of its capabilities.

REVIEWERS' COMMENTS

Reviewer #1 (Remarks to the Author):

In this revision, the authors have clarified the relevance of the wafer scale processing and better acknowledged the fundamental limitations of their processing. I have no further remarks on the technical aspect of the work and I am content if the manuscript is published in Nature Communications.

A point-by-point response to the reviews of the manuscript “A heterogeneously integrated lithium niobate-on-silicon nitride photonic platform”

First of all, we thank the reviewers for their attentive reading of the new version of our manuscript. We appreciate the comments given by both referees and we give an explicit reply in this letter. Following the reviewers' questions, we made several changes to the manuscript that we believe will help to better convey the novelty of our work. For convenience, the referees' comments are colored black while our responses use blue, and changes to the manuscript are colored red both here and in the manuscript file.

Referee #1:

I appreciate the efforts of the authors in improving the transition losses by fabricating the etched tapers in lithium niobate. However, based on the authors' replies, I believe they might have misunderstood the main concerns from my earlier report, namely the low conceptual novelty with respect to the state of the art, the rationale of the approach, and the limited performance of the modulators.

We thank Reviewer #1 for his/her positive feedback on the transition losses. We also agree that the question of novelty should be addressed properly, as in the previous reply letter we focused more on the reviewer's technical remarks. In this reply letter, we will focus only on the novelty issues raised by the reviewer and we hope we will be able to meet his/her expectations.

1. While it is clear that this is the first time that a LN wafer is bonded on silicon nitride photonic integrated circuits, it remains unclear whether this approach provides important advantages with respect to die bonding (earlier works) given that the latter: a) also enables integration at the wafer level; b) is easier to implement as it has relaxed requirements in terms of wafer homogeneity; c) optimizes the real state of an expensive wafer. The tradeoffs between these approaches should be better conveyed and acknowledged in the introduction.

The question of the advantages and disadvantages of the wafer-level bonding approach presented in our work is indeed an important one to be discussed in the manuscript. Die-level bonding can be advantageous in some cases, as it was stated by the Reviewer. We cite here a related work from the field of MEMS and semiconductor processing, discussing the question of choice between wafer-wafer stacking and chip-wafer/chip-chip stacking integration:

T. Matthias et al., “3D Process Integration – Wafer-to-Wafer and Chip-to-Wafer Bonding”, MRS Online Proceedings Library, **970**, 408 (2006)

“Wafer-level integration has the advantage of higher throughput, enhanced cleanliness and the flexibility that standard fab equipment can be used for further processing. 3D integration applying chip-to-wafer bonding focuses on the yield (“good known die”) and enables to stack dies of different size e.g. several small dies on one big base die.”

In other words, the wafer-level bonding is not preferable in cases, where the bonding yield is not good enough to risk the whole wafer, or if the acceptor wafer needs complicated processing. In these cases, as it was stated by the Reviewer, the die-level approach indeed “optimizes the real state of an expensive wafer”. However, in our case of LNOI wafer bonding, we treat the acceptor wafer with **standard cleanroom techniques** (CMP, ALD, etc., see SI section I for details). Moreover, the **bonding yield** of our process is excellent with micrometer-scale de-bonded areas at the edges of the wafer as we state in the manuscript. This is also proved by the fact that the devices we measure across the wafer perform the same way.

We also cite this paper in our revised version of the manuscript and put the following sentences in the introduction:

Compared to die-level integration, wafer-scale bonding has several distinct advantages, chief among them a significantly increased wafer throughput that is advantageous in high volume applications [52]. In addition the approach is also extendable to larger wafer size (e.g. 6 or 8 inch) and enables further processing using standard fabrication techniques, such as DUV lithography used in this work for the adiabatic transitions fabrication and metal lift-off. Given the abundance of LiNbO_3 in contrast to III-V materials, widely used in die bonding, and the availability of LNOI in large wafer sizes, the wafer-level integration becomes an attractive and cost effective method for heterogeneous LNOI PICs.

2. Regrettably, my earlier comment “The electro-optic modulator has an extremely large $V_{\pi L}$. This value is not only significantly higher than ridge LN waveguides (which is expected) but also higher than the results presented in ref. 45 (6.67 V-cm) that relies on a hybrid waveguide that combines bonding of LN on a processed silicon nitride circuit. The authors argue that the $V_{\pi L}$ can be improved with an optimized design of the core (cf. Fig. 3 in S3), but this results into a fundamental tradeoff with optical field confinement and, I suspect, the ability to achieve broadband dispersion engineering of the hybrid mode which is a key property enabled by the thick Si_3N_4 waveguides fabricated with the Damascene process.” is not well addressed.

The authors do acknowledge that their modulators display lower performance, and they mention in their reply that “The critical aspect of our work is not the electro-optic efficiency itself but the combination of low optical losses at wafer-scale (< 0.1 dB/cm and 2.5 dB/facet insertion loss) with electro-optics ($V_{\pi L} \approx 30$ Vcm) that can be optimized down to approximately 8 Vcm without the need to etch lithium niobate, therefore preserving the low optical losses”. However, it remains unclear whether the optimization process (which essentially relies on modifying the aspect ratio of the silicon nitride waveguide to enhance the confinement in the LN slab) can indeed preserve the low optical losses, keep the transition losses low and attain dispersion engineering.

We apologize for being not so clear in our previous reply, but we draw the reviewer’s attention to the fact that we demonstrate phase modulators having 8.8 Vcm and 5.6 Vcm $V_{\pi L}$ values. As stated in the text of the manuscript, we observe insertion loss increase for the highest participation case (52%), while for the 8.8 Vcm configuration we see no additional loss due to the close proximity of electrodes to the optical waveguide. Therefore, we indicate 8.8 Vcm as our main $V_{\pi L}$ value metric. Those values are not simulations, but measurements (which correspond well to the

simulations). It appears that in the previous reply letter, we did not update the $V_{\pi}L$ metric in the comparison table. We put the latest version of the comparison table of the manuscript (Supplementary Table 1) here:

Reference	Intrinsic Q-factors	Linear optical loss	$V_{\pi}L$ product (Tuning rate)	Insertion loss	Wafer-level fabrication	Statistical analysis
This work	$4 \cdot 10^6$	0.1 dB/cm	8.8 V·cm (42 MHz·V ⁻¹)	2.5 dB/facet	Yes	Yes
[13]	10^7	0.027 dB/cm	No data	No data	Yes	No
[14]	No data	No data	13.4 V·cm	6.5 dB/facet	No	No
[15]	$2.5 \cdot 10^6$	No data	(500 MHz·V ⁻¹)	No data	Yes	No
[16]	No data	0.2 ± 0.4 dB/cm	No data	5 dB/facet	No	No
[17]	$1.8 \cdot 10^6$	0.27 dB/cm	No data	1.7 dB/facet ^b	Yes	Yes
[18]	$7.68 \cdot 10^5$	0.2 dB/cm	5.1 V·cm ^a	6.5 dB/facet	Yes	No
[19]	No data	No data	6.2 V·cm	No data	Yes	No
[20]	No data	7 dB/cm	6 V·cm	>10 dB/facet	Yes	No

We also modify Figure 1 panel (h) to demonstrate the measured $V_{\pi}L$ metric explicitly to avoid any confusion for the electro-optic performance:

Figure 1: ... (h) Half-wave voltage measurements for phase modulators with 4 mm length using Mach-Zehnder interferometer. (i) FEM simulations of hybrid optical mode profile for a waveguide with 38% optical mode participation in lithium niobate and 6 μm separation gap between electrodes (see SI for electro-optic simulations).

As indicated in the manuscript text, these measurements are done on the “optimized” silicon nitride waveguides on a separate design run with 600 nm waveguide thickness.

The question of additional optical losses, transition loss, and dispersion engineering of the “optimized” structure is indeed important to be addressed further.

- 1) On the **linear optical loss**. In the last version of our manuscript, we added simulation results of the bending loss for high-confinement waveguide configurations (SI section V). Unfortunately, on the electro-optic design run (D88_12), we have much less optical loss data, as the main aim of the design was to achieve efficient electro-optic modulation, in contrast to the low-confinement design (D67b_01), where we explicitly study optical losses on different microresonator configurations. In the revised version of the manuscript we add linewidth measurements of 2 different microresonators in the “optimized” waveguide configuration with 2 different bending radii.

To verify the model, we measure Q-factors of two microresonators in the electro-optic waveguide configuration: 120 μm and 180 μm radii (Figure S5(c-d)). The spacer thickness is around 80 nm according to the cross-section SEM measurements, and waveguide width is set to be 2 μm for both resonators. The smaller ring reveals an increase in the resonance linewidth, which leads to the Q-factor of $3.9 \cdot 10^5$, while the larger ring has almost a magnitude higher Q-factor ($2.2 \cdot 10^6$), close to the ultimate measured values presented in the main manuscript. The latter means that for 80 nm spacer thickness the loss is not dominated by bending already at 180 μm radius. The “waving” background in panel (c) corresponds to the standing waves inside the bus waveguide and are visible due to the 16-GHz scale of the frequency axis (the total resonance linewidth in this case reaches approximately 800 MHz). We also mark these measurement results with red crosses on the chart presented in Figure S5(a).

Supplementary Figure 5: Bending loss-limited Q-factors in hybrid waveguides. (a) Simulation results for the “electro-optic” waveguide geometry for different bending radii and spacer thicknesses. Crosses correspond to the measurements presented in panels (c-d). (b) Example of an optical mode dissipation in case of a thick spacer. The inner waveguide colors display linear-scale mode distribution, while the far-eld lines are in log scale to visualize an optical mode dissipation in the tail, being de-coupled from the Si_3N_4 waveguide core. (c) Resonance fitting (red curve) of a 120 μm microresonator in the simulated configuration with 80 nm spacer thickness. (d) The same measurement for a resonator with 180 μm radius. Note the frequency scale change for panels (c)-(d).

- 2) On the **transition loss** (fiber-to-chip coupling). The transition loss results we report in the last version of our manuscript, as well as all the measurements displayed in Figure 3, are measured on the high-confinement configuration of the waveguide. In other words, those are results from wafer D88_12. In the revised text we explicitly mention this fact to give our readers a better understanding of which configuration was used in the experiment.

Note that the presented results are achieved on the high-confinement waveguide configuration (600 nm Si₃N₄ thickness).

Please also note that the fiber-to-fiber transmission measurements display not only the transition loss but also include the linear loss in the 5 mm long waveguides in the electro-optically “optimized” aspect ratio configuration. The total insertion loss does not exceed 5 dB fiber-to-fiber, most of which comes from the fiber-to-Si₃N₄ waveguide coupling as we discuss in the manuscript. However, as we stated before, we find the linear loss study based on the fiber-to-fiber transmission measurements not reliable and not accurate enough for the PIC characterization, so we did not include these discussions in the manuscript to not confuse our readers.

- 3) On the dispersion engineering. We thank Reviewer 1 for this question as our initial version lacked the discussion on dispersion engineering for the high-confinement waveguides. If we consider dispersion engineering as the ability to invert dispersion from normal to anomalous by changing the waveguide width, the “optimized” structure indeed lacks it. In the high-confinement regime (waveguide thickness 600 nm) we do see only normal dispersion for the silicon nitride waveguide width range of 0.8-2.0 μm. To make this fact clear for the readers interested in this aspect, we add the following sentence in the revised version next to the dispersion engineering discussions:

The presented design demonstrates an example of devices that are uniformly coupled over a broad frequency range and, at the same time, experience flat integrated dispersion with $D^2/2\pi$ of $O(100 \text{ kHz})$. Note, that the flat and anomalous dispersion can be achieved for the low-confinement waveguide configuration (950 nm waveguide thickness), while for the electro-optic configuration (600 nm waveguide thickness) the dispersion is strongly normal for any waveguide width. The trade-off between dispersion engineering and electro-optic interaction strength is a question of specific designs for specific experiments.

Despite the fact, that the near-zero dispersion control is achieved only for the low-confinement regime, we believe that our platform still remains versatile for many applications, as one can choose between the LN confinement factor and the ability to achieve anomalous dispersion depending on the specific experiment.

Reviewer #2 (Remarks to the Author):

I appreciate that the authors responded in detail to my previous comments. The

paper is reorganized and has toned down many of its original claims. The authors, for the most part in the rebuttal, acknowledged that my original three major concerns are valid: 1) The novelty of the platform; 2) CMOS compatibility; 3) The performance.

We thank Reviewer #2 for his/her positive feedback on the text improvements.

The question remains – what makes this paper stand out compared to the references 47 (Boynton et al., Optics Express 2020), and 48 (Chang et al, Optics Letter 2017)? I agree with the authors that, as a platform, this work demonstrates potential for building complex circuits using wafer-scale PDKs. There are advances at various levels in each demonstrated building block. However, these blocks remain components and do not enable complex functional circuits.

As we state in the response to Reviewer #1, the reliable wafer-scale bonding approach helps to increase the **process throughput**, which is one of the key metrics for the future widespread of integrated lithium niobate photonics. We also improve other metrics compared to the references cited by the Referee, mainly **the optical loss**. We also demonstrate the performance of our platform in **various applications**, which was not shown for heterogeneously integrated LNOI approaches before. As we discussed in the previous reply letter, we believe that the demonstration of complex functional circuits sets the aim for future works, aimed at specific experiments. In the current manuscript, we demonstrate the basic performance of our platform with an extensive analysis of its capabilities.